# The policy effect of green finance reform and innovations: Empirical evidence at the firm level

**Hanghang Dong**◉*⊕, **Miaomiao Tao**⊕

Department of Economics and Applied Statistics, Faculty of Business and Economics, Universiti Malaya, Kuala Lumpur, Malaysia

⊕ These authors contributed equally to this work.
* donghanghang1991@gmail.com

## Abstract

The Chinese central government established eight pilot zones in five provinces for green finance reform and innovations (GFRI) in 2017. The pilot zones promote green finance development and explore the propagable and reproducible experiences regarding mechanisms and institutions. Adopting a sample of China's listed companies from 2012 to 2021, this paper constructed a quasi-natural experiment and investigated the GFRI policy's effect on firms' total factor productivity (TFP) using the difference-in-differences (DID) method to verify the implementation effect of the GFRI policy. Furthermore, heterogeneity analysis and mechanism analysis were conducted to identify the guidance effect and deep mechanisms of the GFRI policy. The empirical results demonstrated that firms' TFP in pilot zones increased substantially after implementing the GFRI pilot policy, confirming that the policy had a strong incentive effect. The corresponding promoting effect was particularly significant for non-state-owned companies, the eastern and central regions, and firms in the growth stage. Further mechanism analysis revealed that the GFRI pilot policy can stimulated firms' TFP by promoting technological innovation and improving resource allocation efficiency. This paper's empirical findings are essential in improving relevant policies and expanding the pilot zones.

## 1. Introduction

As the world's largest developing country, China has achieved remarkable economic growth since implementing its forward-looking policies of reform and opening up. China's traditional economic development pattern, characterised by the massive investment of social capital and human resources, has curbed long-term sustainable and inclusive economic development and triggered serious environmental issues [1]. This situation has made China facing a critical issue to overcome these challenges [2]. Against such a developmental background, green finance, a useful environmental instrument with dual characteristics of environmental concerns and financial support, has been proposed to balance the relationship between economic development and environmental protection by the Chinese government [3]. Steered by the

**Funding:** The authors received no specific funding for this work.

**Competing interests:** The authors have declared that no competing interests exist.

new development ideology of "lucid waters and lush mountains are invaluable assets", the Chinese government has released a series of policies to facilitate the construction of an environmentally-friendly society. Among these, establishing pilot zones of GFRI acts as an essential component. The executive conference of China's State Council selected eight pilot regions attempting to establish the first batch of pilot regions for GFRI on 14[th] June, 2017. Furthermore, five main tasks including green financial instrument, green finance platform, financial service institutions, government support, and risk management, were put forward in this meeting, which provided a clear picture for establishing pilot zones for GFRI in China.

With the improvement of related policies, green finance has also received extensive attention from adademia in recent years [4]. The Ref. [5] noted that green finance represented a new financial instrument that integrates environmental protection with that of economic benefits, highlighting "green" and "finance" had become two controversial issues. While as a concrete form of environmental regulation in the financial market, green finance can play an essential role in accelerating the transformation to sustainability [6]. It can also be viewed as a powerful weapon to address environmental problems. According to the Ref. [7], the core of green finance strengthens the financial ability to enhance environmental quality further. Researchers have, until now, focused more on fundamental matters regarding green finance in promoting environmental protections. These matters have included; establishing a green finance system within environmental protection systems [8], how to effectively incorporate the environmental protection into green finance [9], and the effect of policy implementations related to green finance [10]. Some studies have also been relied on empirical works relevant to the correlation between green finance and the environment. For instance, the Ref. [11] found that green finance significantly reduced $CO_2$ emissions, while the effects of green finance policies continually fell short and lacked continuity. Still, the Ref. [12] demonstrated that public spending on green projects significantly contributed to green growth.

Pioneering studies have analyzed the effect of green finance by constructing comprehensive indexes. They have also pointed out that green finance helps promote; sustainable energy development, sustainable economic development [13], enhanced environmental quality [14], and reduced carbon emissions [11]. Furthermore, several studies have examined the effects of individual green financial instruments and policies. For example, the Ref. [15] examined the effect of green credit on carbon emissions reduction in China and further noted that green credit was conducive to reducing carbon emissions in China, while the corresponding effect had significant regional heterogeneity. The Ref. [16] explored the effects of the green credit policy from the provincial and national aspects and concluded that the implementation efficiency level was fall far short of expectations. By contrast, the Ref. [17] highlighted that green credit was crucial in promoting industrial structure adjustment. As an essential tool to divert financial resources to inclusive development, green bonds can facilitate green economic development without penalising the issuers financially [18]. Similarly, the positive effect of green bonds can also be detected in the work of the Ref. [19], who concluded that green bonds had a positive effect on stimulating GDP growth. It was also worth noting that the studies cited above have mainly focused on the macro level, making it challenging to understand the mechanism of green finance, and it is also difficult to provide more practical recommendations at the micro level.

Promoting the green and low-carbon transformation of firms is a crucial criterion for evaluating the impact of green finance. Thus, several studies have been conducted by adopting micro-level dataset. Specifically, green credit can significantly strengthen the innovative capacity of environmental protection firms [20] and can accelerate green innovations by easing financial constraints [21]. Albeit with the environmental protection function, green finance can also affect the capital allocation efficiency of financial institutions and banks, thereby

influencing the financial performance of firms [22]. In this regard, many studies have explored the effects of environmental regulation policies such as environmental taxes or emissions trading on corporate debt financing [21, 23]. However, few studies have paid much attention to whether green finance and related policies, rather than financial tools (i.e., green credit and green bonds), have achieved considerable intention to promote the green transformation of firms in China. Environmental issues are mainly caused by enterprises pursuing huge profits, which has led to high energy consumption and polluent emissions [13]. In this regard, curbing the expansion of heavily-polluting firms and increasing incentives for investment in research and development (R&D) activities, are two fundamental paths to achieving green transformation and pollution reduction in China at the micro level. Therefore, this study has mainly centered on the impact of green finance on firms' productivity and its heterogeneous effects. It has further revealed the transmission mechanisms by which green finance affects firms' productivity in order to make firms (especially for heavily polluting firms) change their production behaviour. Thus, it has provided new insights into modelling the effects of China's green finance pilot policy.

Thus, the marginal contributions of this paper can be summarised as follows. First, again, the existing studies [24–27] have paid much attention to investigating the effect of green credit but have ignored the role of green finance in firms' TFP. While to the best of this study's knowledge, this paper has provided the first evidence to demonstrate the promoting effect of the pilot policy for GFRI on firms' TFP stimulation by adopting a quasi-natural experiment. This research may lay the empirical foundations to support further analysis. Second, in the heterogeneity analysis, this study not only captured heterogeneity effects based on firms' attributes (i.e., SOEs and non-SOEs) and regional levels (i.e., eastern, central, and western regions), but also considered firms' different growth paths (i.e., growth type, mature type, and declining types). Thus this study has provided an integrated scenario. Third, enlightened by the Porter Hypothesis [28], green finance as a useful environmental regulation instrument could encourage firms to invest more in innovation activities. However, existing studies [29, 30] ignored this aspect. Still, this study also explored how the green finance pilot policy affected firms' TFP through the channel of energy efficiency utilisation. This exploration was because politically connected firms and stated-owned enterprises (SOEs) may easily acquire financial resources. After all, their stipulated environmental constraints are relatively weak [31]. In this case, if financial resources are channeled into firms with lower marginal productivity, it could result in resource misallocation issues and eventually hinder TFP improvement.

This paper proceeds as follows: Section 2 describes the concepts of green finance, introduces the institutional background, and then proposes the theoretical hypotheses. Section 3 includes; the model specification, selection of variables, and data descriptions. Section 4 reports the empirical results and performs a set of robustness tests. The heterogeneity analysis will be conducted in Section 5. Section 6 performs a mechanism analysis, while the related research conclusions and the corresponding policy implications are included in Section 7.

## 2. Literature review

### 2.1 Concepts of green finance

Green finance has no single, complete, unified, or explicit definition, even though it is increasingly used globally [32]. However, the existing literature has clarified the concepts of green finance from three crucial perspectives. Firstly, green finance refers to financial innovations designed to manage natural environmental risks through various; financial initiatives, processes, services, or products (i.e., green bonds, green stock, and green credit) [33]. Secondly, green finance has been considered environmental finance [34] that aims to provide financial

derivatives and investments to achieve environmental protection [35–38]. The last group highlights that green finance as a financial service supporting environment-friendly investments and provides sufficient financial resources for green-oriented projects through credit, insurance, securities, and carbon finance to deal with climate change challenges [39–41]. Although the concept of green finance has been heterogeneous across different studies, green finance has its official definition in China's Guidance on Building a Green Financial System in 2016. More specifically, green finance means economic activities dealing with climate changes (The logic for this path analysis lies in the policy. The pilot policy entitled "green finance reform and innovations" focuses on green finance development and technology innovations. Hence, this study believes that such a policy may also play a crucial role in firms' technological progress), maintaining environmental conditions and utilising resources efficiently. In particular, green finance can provide financial services for; project operation, project financing, renewable energy saving, green transpiration, risk management and environment protection. Therefore, clarifying the concepts of green finance plays a fundamental role in exploring the effects of green finance and its associated policies.

## 2.2 Green finance and TFP

Some controversy exists regarding the effects of green finance on firm-level TFP. The compliance cost theory indicates that; R&D, process-upgrading, and equipment purchasing, requires a huge amount of material costs, yet green finance can raise the financial constraints for "two-high" firms (i.e., high energy consumption and high $CO_2$ emissions). Therefore, firms must take on higher risks and additional costs [39, 42]. Consequently, firms may reduce their R&D, investment activities, and production inputs, thereby hindering productivity improvement [43]. Consistent with the compliance cost theory, the Ref. [44, 45] highlighted that strict environmental regulation negatively affected firms' productivity. This outcome was because the additional costs incurred by firms have exceeded the benefits they could obtain, and thus, negative influence of environmental regulations on firms' TFP was identified [46, 47].

Unlike the compliance cost theory, the Porter Hypothesis asserts that well-designed environmental regulations can force firms to conduct R&D activities, strengthening commercial competitiveness and creating a win-win situation between the environment and the economy [48]. Consistent with the Porter Hypothesis, the Ref. [49] have demonstrated that a properly designed environmental regulation policy positively effected productivity in Canada. Similarly, environmental regulation was proved by [50] to significantly contribute to firms' TFP through the channel of technological innovation. Moreover, a strict and standardised environmental regulations can motivate firms to adopt energy-saving production equipment, which is expected to improve their productivity.

Innovation, green, openness, coordination, and sharing comprise the "Five New Development Concepts", which were proposed in 2015 at the Fifth Plenary Session of the 18th CPC Central Committee, emphasising the importance of sustainable development. Thus, ensuring a suitable and reasonable environmental regulation policy is essential for realising sustainable development. Policies of environmental regulation can be categorised into two kinds. One refers to common-and-control regulations, which are characterised by administrative supervision and control (i.e., Air Pollution Prevention and Control Action Plan); the other mainly regulates the market-based environmental regulations. The government has given the market a decisive role in pollution mitigation and resource allocation (i.e., China's carbon trading market). Unfortunately, the Ref. [51] asserted that China's current environmental regulation policies had not achieved the prescribed reduction target. Still, empirical findings provided by the Ref. [52] showed that the inhibitory effect of the market-based environmental tools on

pollution reduction was relatively weak. At the same time, if the efficient market hypothesis is invalid in China, it is also hard to realise energy savings and emission reductions.

Finance has been treated as a fundamental lever to support sustainable economic development aimed at environmental regulation [53]. In China, financial resources have been mainly channelled into capital-intensive key industrial sectors with high energy consumption. This situation has resulted in environmental issues and overcapacity [54]. On the contrary, green finance can guide the flow of social and financial capital to sustainable projects to promote cleaner production [23]. In the meantime, it can also promote the optimal redistribution of other production factors. Hence, vigorously designing and introducing green financial policies to address climate change and environmental issues is beneficial and give full play to finance's resource reallocation.

## 2.3 Institutional background

Revisiting China's green finance progress, in 2007, green credit was first introduced, and formally enacted in 2012 by the Chinese central government as a financial instrument to effectively deal with environmental issues [23]. The policy was not significantly promoted due to a lack of policy details and evaluation standards [16]. To this end, the National 13th Five-Year Plan further mentioned that China would facilitate establishing a green financial system that subsequently was jointly released by seven official departments in 2016, serving as the top-level green finance initiatiative in China [13]. Thus, the framework of green finance in China was formulated, and green finance became a powerful environmental regulation instrument in China.

Although green finance has developed quickly in China [14], its overall performance has not been high-quality [55], and there has been insufficient experience because of the late start. In this regard, the Chinese government has strongly suggested that it was necessary to set up several pilot zones for green finance reform to explore reproducible experiences and refine green financial systems. Therefore, in 2017, the Chinese government approved eight pilot zones in five provinces for green finance reform and innovations, which meant that China's green finance had entered a new developmental stage of a combination of top-level design and bottom-up practice.

It is noteworthy that the selected pilot zones in the five provinces are represented economic strength and resource endowments, which reflected the representativeness and differences of the zones. The selected eights zones can be separated into three categories based on their different functions. Guangdong Province and Zhejiang Provinces have more advanced financial systems; thus, they were classified into the first group. Two cities in Zhejiang Province (i.e., Quzhou and Huzhou) have been regulated to investigate the realisation mechanism of the conviction that "lucid waters and lush mountains are invaluable assets" in financial fields and innovative services in green finance. Guangzhou in Guangdong Province mainly in charge of developing the financial market. The financial systems in Guizhou Province and Jiangxi Provinces are relatively weak. Hence, these two provinces have been classified into the second group. They have been advised to develop new green development patterns through using high-quality green resources. The third group comprises three zones in Xinjiang Province. They have been tasked to develop green finance to support modern agriculture development vigorously, thus, fully performing green finance's outward radiating and demonstration role in constructing the Green Silk Road.

One year after the GFRI pilot policy implementation, more than 85% of tasks in the whole framework had been successfully promoted. Reported by incomplete statistics, green loans balance in the selected pilot zones has amounted to approximately 8.23 trillion Yuan by the end

of 2018. Still, the same statistics indicated a balance of more than RMB 236.83 billion Yuan by the end of 2020, with a sharp increase of about 15.1% over the initial implementation level.

## 2.4 Hypothesis formulation

It is noteworthy that green finance can raise the loan threshold for companies through financial constraints [44]. Specifically, due to the financial institutions and banks' ability to control the direction of financial resources, they can provide financial resources for those "two-high" firms (i.e., a relatively high interest rate for "two-high" firms). On the contrary, they can stratify the need for financial supports for environmentally-friendly firms. In this case, there are two implicit impacts. On the one hand, when financial resources are constrained, the production capacity of "two-high" firms is reduced significantly; energy consumption and pollution emissions are also reduced simultaneously.

On the other hand, because the total amount of financial resources is limited in a given period, sufficient financial resources will be channelled into energy-intensive firms. Subsequently, they are able to invest more in R&D activities, especially for green technologies, thereby contributing to their productivity. Under these circumstances, firms are expected to develop new technological infrastructure, so as to achieve green transformation, upgrade their productivity, and obtain competitiveness and prestige [55, 56]. Thus, this study proposes its first hypothesis. Moreover, Fig 1. displays the theoretical notion where how green finance can affect firms' productivity.

**Hypothesis 1.** The implementation of the GFRI significantly improves firms' TFP.

Green finance especially through financial institutions that plays a crucial role, by limiting financing firms' access to financial resources (i.e., green credit or green bonds) [57]. Thus, the

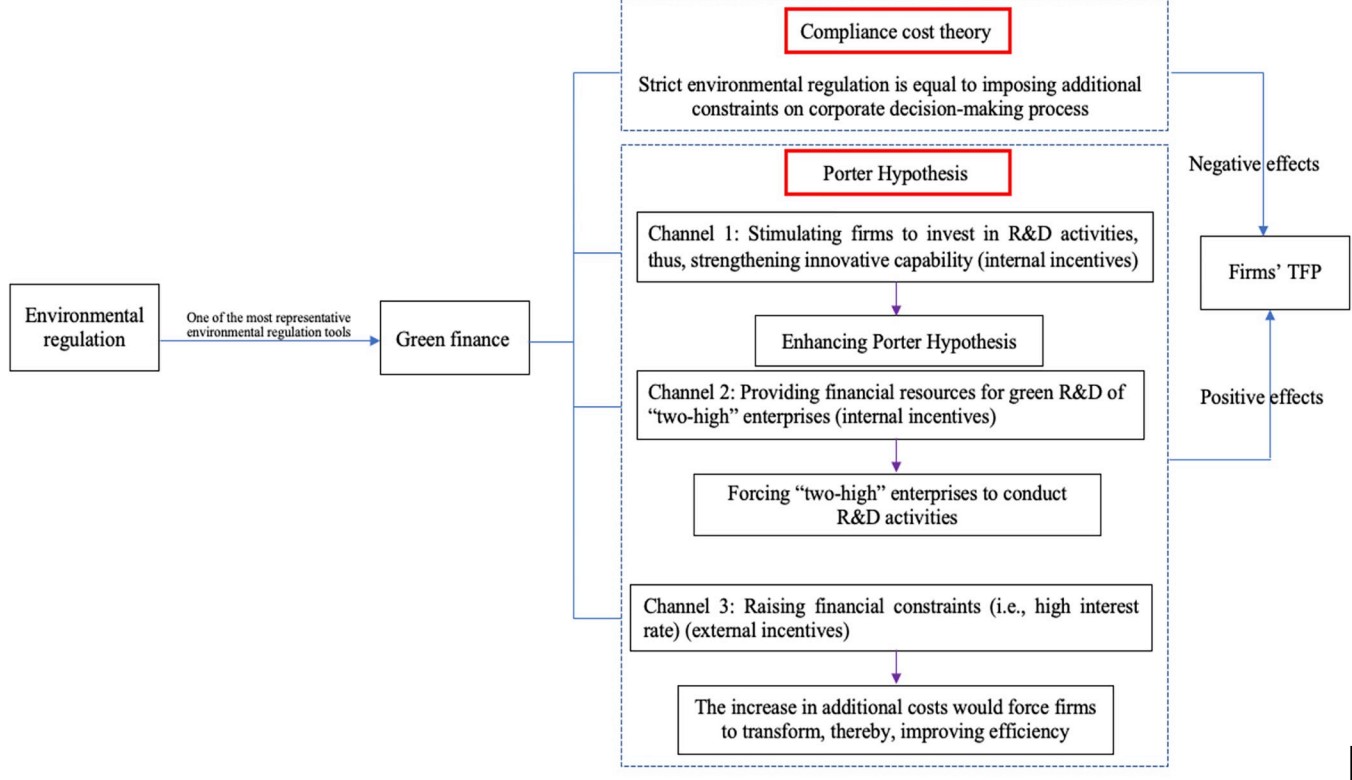

**Fig 1. Theoretical notion.**

impact of green finance on TFP varies between firms, with different external monitoring mechanisms, firm ownerships, and regions [58].

Many Chinese firms have a high level of state ownership, offering them easier access to resources and other privileges. Thus, SOEs may receive more compensation from the government than non-SOEs [59]. Additionally, the state-owned banks dominate the banking system, thus, leading to reduced supervision over SOEs compared to non-SOEs that face strong banking supervision and financing constraints, making it easier for SOEs to access bank loans [60]. Thus, it is believed that non-SOEs face significantly greater pressure compared to SOEs when confronted with the same green finance policy shock.

**Hypothesis 2.** The effect of the GFRI on firms' TFP is more obvious in non-SOEs than in SOEs.

There are obvious distinctions in the level of economic development, the accumulation of human capital, and the level of industrialisation in the eastern, central and western regions of China. Hence, the different degrees of government intervention, financial development, and resource allocation across China's three major economic zones may behave differently in the face of green finance policy shocks. Specifically, since the overall administrative efficiency and level of education in eastern China are higher than those of the central and western regions, eastern China also has stronger environmental awareness. Moreover, higher regional innovation also provides a good environment for implementing the GFRI policy. Higher government efficiency will help firms invest more in technological innovation and [61]. Thus, the promotional effect of the pilot policy for the GFRI policy in the eastern region is more significant compared with the central and western regions.

**Hypothesis 3.** The effect of the GFRI on firms' TFP is more significant in companies located in eastern China.

Under environmental constraints (i.e., huge penalties, large pollution taxes, production suspension and closure), firms, especially "two-high" firms, will redesign their investment strategies. Such changed investment strategies are because they must reconsider their market development prospects and future financial environment. Thus, after implementing of the pilot policy for CFRI, the compliance costs encountered by the firms may increase, and the pressure for survival could be aggravated. Firms are motivated to improve their production technology and reduce local pollutant emissions to realise long-term sustainable development, thus, choosing to enhance technological innovation [17]. Technological innovation will accelerate the increase of firms' non-patented and patented technologies, enhancing firms' knowledge stock. Firms' knowledge stock may eventually be transformed into production capacity and stimulate firms' productivity through a set of intermediary factors [62, 63]. As mentioned earlier, under strict environmental regulations, "two-high" firms will be phased out of the market; energy-saving firms, therefore, will obtain sufficient financial resources to strengthen their innovative capabilities, resulting in overall efficiency improvement. Higher resource allocation efficiency will ensure that economies of scale are well utilised and improve firms' productivity [64].

**Hypothesis 4.** The GFRI improves firms' TFP by promoting technological innovation and resource allocation efficiency.

The firms' life cycle theory suggests that firms evolve through distinct stages in its life cycle. Profitability, revenue generation, and cash flows are uncertain during the introduction and growth stages. Such growth firms face a "liability of newness" and are prone to exit the market [65]. Due to intense competition, growth-type firms mainly focus on product improvement and modification. In contrast, decling-type firms pay more attention to survival. Therefore, firms in the introduction, growth, and declining stages show "fragile financial performance", which may jeopardise shareholders value [66]. However, revenue growth is relatively stable;

cash holdings are relatively sufficient in mature-type firms, and, thus, overall risks and uncertainties are relatively lower.

Additionally, the Ref. [67] further pointed out that the costs and paths of accessing information and financial resources varied for firms at different developmental stages. Still, compared with mature and declining type firms, with growth-type firms encountered greater information risks because of; incomplete information discourse, higher information asymmetries, and higher information costs. Moreover, they also faced higher capital needs and higher financial constraints [68].

**Hypothesis 5.** The promoting effect of GFRI on TFP stimulates growth-type firms more than mature and declining firm types.

## 3. Methodology

### 3.1 Model specification

As a commonly implemented econometric procedure in policy evaluation [69, 70], the purpose of the difference-in-differences (DID) method is to examine the treatment effect of a specific public policy. The whole sample is naturally classified into two groups, namely, a treatment group and a control group, and two periods: the "before" and "after" periods. More specifically, the DID method includes two essential procedures. The first difference regulates the average difference of the outcome variable of the two groups that will remove unobserved group heterogeneities that do not change over time. Conditional on the first difference, the last difference is the difference between the "before-after" period, which removes the common trend contributed by the two groups. The DID method design served as a useful framework for this paper. This is because the GFRI's implementation can be considered as a quasi-natural experiment due to the insufficient expectation prior to its announcement. For simplicity, the GFRI's implementation can be viewed as an exogenous intervention for the subject. More specifically, the DID model in this paper was constructed as follows:

$$y_{it} = \alpha_0 + \alpha_1 \underbrace{treated_{it} * post_{it}}_{did_{it}} + \beta \sum X_{it} + \varepsilon_{it} \qquad (1)$$

Where $y_{it}$ is the explained variable, which regulates the firms' total factor productivity in firm $i$ at time $t$; $treated_{it}$ is a sorting dummy variable. If firm $i$ belongs to the treatment group, then $treated_{it} = 1$; 0 otherwise. $post_{it}$ represents a time dummy variable for the implementation of the pilot policy for the GFRI. Before the GFRI pilot policy implementation, $post_{it} = 0$, and after the policy implementation, $post_{it}$ is 1. $treated_{it}*post_{it}$ is the interaction term between the group dummy variable $treated_{it}$ and the pilot policy's implementation dummy variable $post_{it}$. The coefficient $\alpha_1$ captures the net effect of the GFRI pilot policy. If $\alpha_1$ is significant at a certain statistical level. This situation implies that implementing the pilot policy for the GFRI has a significant effect on firm-level TFP. $X_{it}$ denotes a set of selected control variables, $\beta$ is the corresponding response parameter; $\alpha_0$ is the intercept, while $\varepsilon_{it}$ is the error term.

### 3.2 Selection of variables

**3.2.1 Firms' TFP.** The OP method [71] and LP methods [72] have been widely adopted by the existing studies to measure firm-level TFP [73, 74]. Essentially, the OP method uses a firm's exsiting investment as a proxy variable for the productivity influence, while requiring the relationship between the investment and the total output always to remain monotonous, meaning that it is not possible to measure TFP with zero investment. This method omits a lot of samples because not every firm has a positive investment every year. The LP method offers a

new strategy to measure a firm's TFP by treating the intermediate inputs as proxy variables instead of the firm's investment. Thus, this research estimated firms' TFP by performing the LP method. The firms' TFP estimated by the OP method was also reported simultaneously to ensure the robustness of the empirical findings.

The LP method was carried out based on the Cobb Douglas production function:

$$y_{it} = \gamma_0 + \gamma_l L_{it} + \gamma_k K_{it} + \varpi_{it} + \epsilon_{it} \tag{2}$$

Where $y_{it}$ represents the logarithm of firm $i$'s total output at time $t$, which is proxied by value-added. $L_{it}$ is the logarithm for the freely variable labour of firm $i$ at time $t$. $K_{it}$ is the state variable capital of firm $i$ at time $t$. $\omega_{it}$ and $\epsilon_{it}$ represent two error terms. While the crucial difference is that the former is a state variable which affects the corporate making-decision rules, the latter is random noise and has no potential influence on the corporate decision-making process.

The intermediate input $M_{it}$ is assumed to depend on the firms' state variables $K_{it}$ and $\omega_{it}$, where $M_{it} = (K_{it}, \omega_{it})$. The demand function is monotonically increasing in $\omega_{it}$, thus, the $M_{it}$ function can be inverted, so that $\omega_{it} = \omega(K_{it}, M_{it})$, with $\omega = M^{-1}()$.

This study defines $\varphi_{it}$ as a function of $K_{it}$ and $M_{it}$, as shown in Eq (3):

$$\varphi_{it} = \gamma_0 + \gamma_k K_{it} + \varpi_{it} \tag{3}$$

By substituting Eq (3) into Eq (2), this study achieves the first-stage estimation equation, as regulated in Eq (4):

$$y_{it} = \gamma_l L_{it} + \varphi_{it} + \epsilon_{it} \tag{4}$$

The right side of Eq (4) includes three parts, namely, the linear term $\gamma_l L_{it}$, non-parametric term $\varphi_{it}$, and the error term $\epsilon_{it}$. Specifically, $\varphi_{it}$ can be approximated to a polynomial. Taking $\varphi_{it}$ as an intercept, a consistent estimator can be obtained by regressing Eq (4).

The precondition in the second stage is that a firm's TFP obeys a first-order Markov process, as denoted as in Eq (5):

$$\varpi_{it} = E[\varpi_{it}|\varpi_{it-1}] + \vartheta_{it} \tag{5}$$

Where $\vartheta_{it}$ denotes the innovation in TFP. Therefore, the estimation equation in the second stage is denoted as follows:

$$y_{it} - \hat{\gamma}_l L_{it} = \gamma_0 + \gamma_k ln K_{it} + E[\varpi_{it}|\varpi_{it-1}] + \epsilon^*_{it} \tag{6}$$

Where $\epsilon^*_{it} = \vartheta_{it} + \epsilon_{it}$. A consistent estimator can be obtained because both $\vartheta_{it}$ and $\epsilon_{it}$ are correlated to the state variable $K_{it}$.

Finally, adopting the estimated coefficients $\gamma_l$ and $\gamma_k$, the predicted residual is the logarithm of the firm's TFP by estimating Eq (2).

$$lntfp_{it} = \gamma_0 + \varpi_{it} + \epsilon_{it} \tag{7}$$

The input factors for performing the LP method include the value-added, freely variable labour, intermediate input, and capital. The firm-level value-added was measured by using the income method recorded in the Ref. [75]. The value-added mainly included; labour compensation, business surplus, depreciation of fixed assets, and net production tax. Intermediate input mainly regulates the one-time consumption of raw materials, power, fuel, and other potential physical products and services that the firms purchased from outside, during the reporting period. Intermediate input can be substituted with a proxy variable when considering data availability. Adopting a subset of intermediate input as its proxy variable is reasonable. For instance, Ref. [76] treated electricity power consumption as a proxy variable for intermediate

input. Regarding Refs. [77, 78], the intermediate input was proxied by the cash paid for goods and services.

**3.2.2 Core explanatory variable.**  Currently, China has entered a crucial period of economic development model transformation, and the need for the green finance supporting green industries and inclusive economic and social development is constantly expanding. The transformation seeks to carry out the Opinions of the CPC Central Committee, the State Council on Accelerating the Ecological Civilisation Construction, and the Overall Plan for Reform of the Eco civilisation System. At the same time, it will adhere to the new development concepts of innovative, coordinative, green, openness and sharing, and developing China's green financial market, giving full play to the role of capital markets in optimising resource allocation and serving the real economy. The transformation will further accelerate the construction of ecological civilisation, with the approval of the State Council, the Guiding Opinions of the People's Bank of China, the Ministry of Finance, the National Development and Reform Commission, and other Departments on Building a Green Financial System, released on 31 August 2016, first addressed the significance of establishing a green finance system. In 2017, China's Premier Li, at the State Council executive meeting further announced that China would establish GFRI pilot zones in five regions, namely, Zhejiang, Jiangxi, Guangdong, Guizhou, and Xinjiang (see www.gov.cn), which embarked on the initial exploration of the green finance system in China. It is also noteworthy to point out that two of the selected five pilot zones are located in eastern China (i.e., Zhejiang and Guangdong), one belongs to central China (i.e., Jiangxi), and two comes from western China (i.e., Guizhou and Xinjiang).

Implementating of the pilot policy for the GFRI can be viewed as a quasi-natural experiment. It has been implemented in pilot and non-pilot provinces to control other factors. Thus, the differences in the TFP between the treatment group and control groups before and after the CFRI pilot policy implementation and separated to examine the policy's effect. The core explanatory variable in this paper was $did_{it} = treated_{it}{}^{*}post_{it}$. Specifically, $treated_{it}$ represents a group dummy variable; the firm belonging to the pilot provinces was assigned to 1, and the non-pilot firms were assigned to 0. Since the message was delivered in 2017, *post* is a year dummy variable that equals 1 if the observation year is from 2017 to 2021, and 0 if the observation year is from 2012 to 2016.

**3.3.3 Control variables.**  This study selected a set of control variables used in existing studies to control the possible effects of other factors and obtain robust results. The asset-liability ratio (LR) was used to control the potential impacts of capital structure on firms' TFP [79]. [80] pointed out that large-sized capital-intensive firms had better potential to invest more funds in innovative capability improvement, thus, affecting productivity. Given this, the present study selected the total assets (FS) and the number of employees (EMP) to avoid this bias. Firm age (FA) was also included to avoid the impacts of the life cycle on firms' TFP. [81] highlighted that the expected revenue growth ability (RG) and the stock returns (ROA) are associated with the firm's TFP. This study also introduced CEO duality (DA) as a control variable (DA is a dummy variable. More specifically, DA = 1 when the chairman and general manager are the same one and 0 otherwise). This choice was because the stock options granted to executives significantly and positively impact a firm's efficiency output [82].

The GDP deflator converted all monetary variables into 2012 constant prices. To avoid the outlier effect, all continuous variables were winsorised at 1% level in both tails. The corresponding descriptive statistics of all selected variables are documented in Table 1. It can be seen that the firms' TFP calculated by the LP and OP methods are basically consistent. However, the maximum of 17.929 was nearly 3.669 times the minimum, which meant that firm-level TFP needed further improved.

**Table 1. Descriptive statistics of all variables.**

| Variable | Mean | Std. Dev. | Min | Max | Unit |
|---|---|---|---|---|---|
| TFP_LP | 9.226 | 1.314 | 4.887 | 17.929 | / |
| TFP_OP | 6.587 | 1.073 | 3.287 | 15.183 | / |
| did | 0.120 | 0.325 | 0 | 1 | / |
| LR | 0.431 | 0.216 | 0.032 | 1.376 | % |
| RG | 0.193 | 0.374 | -.0458 | 3.303 | % |
| FS | 22.173 | 1.466 | 18.577 | 27.397 | Yuan (logarithm) |
| FA | 2.06 | 0.946 | 0 | 3.401 | Year (logarithm) |
| DA | 0.296 | 0.456 | 0 | 1 | % |
| ROA | 0.047 | 0.087 | -0.709 | 0.318 | % |
| EMP | 7.581 | 1.321 | 3.091 | 11.379 | Person (logarithm) |

Table 2 presents the Pearson and Spearman correlation matrix. Pearson (Spearman) correlation coefficients between all explanatory variables were quiet low. Further, multicollinearity issues were checked for by cannulating the variance inflation factor. It was found that the largest value was 2.64, which was below the recommended value of 10 [83]. Therefore, there were no multicollinearity problems between the explanatory variables.

Before performing the empirical analysis, this study mapped the average value of firm-level TFP estimated by the LP method, as shown in Fig 2. It can be seen that firm-level TFP has exhibited a significant upward trend, albeit with small fluctuations during the research period. In comparison, the firms' TFP in the treatment group was higher than in the control group. Not surprisingly, the selected five green finance pilot zones were very representative in China. According to the National Bureau of Statistics of China (2022), up till 2021, the total GDP of the above five pilot zones has amounted to 10158.96 billion RMB, accounting for approximately 26% of the national GDP. Therefore, it was acceptable to address that firms' TFP in pilot zones were slightly higher than in non-pilot zones. The firm's TFP in the pilot zones showed a downward trend, suddenly peaking at the bottom, and then exhibiting an upward trend, thus, indicating a "V-shaped" trend during 2018–2021.

**3.3.4 Data sources and preprocessing.** This study collected the dataset of the firms listed on China's Shenzhen and Shanghai Stock Exchange between 2012 and 2021 to explore whether the GFRI's effect on firms' TFP was consistent with the Porter Hypothesis or not. The data

**Table 2. Correlation coefficient matrix.**

| | did | LR | RG | FS | FA | DA | ROA | EMP |
|---|---|---|---|---|---|---|---|---|
| did | 1.000 | -0.015*** | -0.005 | 0.007 | -0.051*** | 0.067*** | 0.029*** | 0.009* |
| LR | -0.018*** | 1.000 | -0.076*** | 0.504*** | 0.381*** | -0.153*** | -0.459*** | 0.364*** |
| RG | -0.020*** | -0.153*** | 1.000 | 0.019*** | -0.347*** | 0.100*** | 0.400*** | 0.019*** |
| FS | 0.002 | 0.491*** | -0.056*** | 1.000 | 0.392*** | -0.198*** | -0.105*** | 0.713*** |
| FA | -0.047*** | 0.382*** | -0.383*** | 0.340*** | 1.000 | -0.245*** | -0.302*** | 0.231*** |
| DA | 0.067*** | -0.150*** | 0.111*** | -0.184*** | -0.243*** | 1.000 | 0.079*** | -0.144*** |
| ROA | -0.022*** | -0.361*** | 0.283*** | 0.004 | -0.213*** | 0.032*** | 1.000 | -0.009* |
| EMP | 0.009*** | 0.329*** | -0.074*** | 0.730*** | 0.214*** | -0.132*** | 0.065*** | 1.000 |

Notes: Table 2 reports the Pearson (below diagonal) and Spearman (upper diagonal)

* $p < 0.10$

** $p < 0.05$

*** $p < 0.01$.

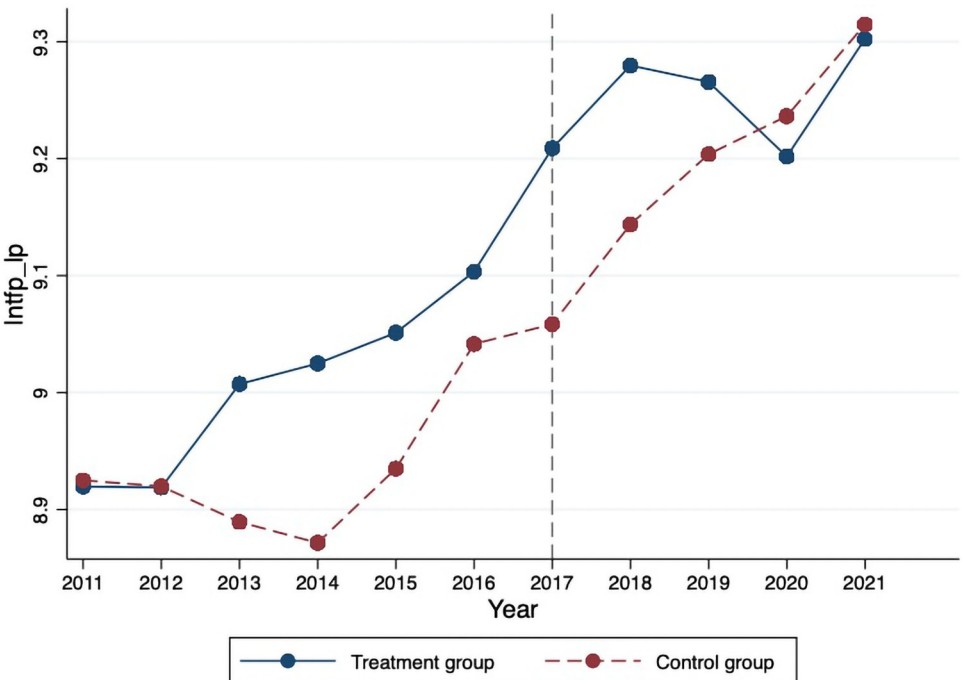

**Fig 2. Mean comparison of firm-level between the treatment and control group.**

were gathered from the China Stock Market & Accounting Research (CSMAR) database and the China Research Data Service Platform (CNRDS). CSMAR database is a comprehensive research-oriented database mainly focus on China Finance and Economy. CSMAR offers data on China stock market and the financial statesments of China's listed companies. Through selection of date range, company code and data fields from more than 4000 tables, specific data can be obtained and exported in Excel (data available at: *https://cn.gtadata.com/*). The selection of control variables was consistent with the pioneering studies, reducing omitted bias. Before conducting empirical analysis, this study dropped firms with missing key variables. At the same time, the inference of outlier values at the 1% level at both tails and clustered all the standard errors of the regression results at the city level were also winsorised. Stata was used to process and analyse the panel data.

## 4. Results and discussions

### 4.1 Baseline results

Table 3 reports the empirical regression results based on Eq (1). Column (1) displays the case without considering the control variables, while Column (2) subsequently includes them. To check the robustness of the result, year, provincial, city, regional (China is divided into three major economic zones according to the classification standards released by the National Bureau of Statistics of China, see http://www.stats.gov.cn/), and industrial fixed effects, as shown in Columns (2)–(5), were gradually introduced. Furthermore, the dependent variable was replaced by TFP_OP in column (7).

Additionally, the coefficients of *did* (in all models) were persistently significant and positive, reaching at least the 1% significance level whether control variables were added or not, which implies that implementing the green finance pilot policy contributed to firms' TFP stimulation as reported in Table 3. The goodness of fit rose from 0.8102 in Column to 0.9156 in Column

**Table 3. Benchmark regression results.**

|  | LP | LP | LP | LP | LP | OP |
|---|---|---|---|---|---|---|
| *did* | 0.1124*** | 0.0597*** | 0.0589*** | 0.0615*** | 0.0622*** | 0.0572*** |
|  | (0.020) | (0.014) | (0.014) | (0.014) | (0.014) | (0.017) |
| LR |  | 0.1839*** | 0.1935*** | 0.2076*** | 0.2064*** | 0.2610*** |
|  |  | (0.050) | (0.048) | (0.047) | (0.047) | (0.056) |
| RG |  | -0.0903*** | -0.0910*** | -0.0941*** | -0.0946*** | -0.1136*** |
|  |  | (0.014) | (0.014) | (0.014) | (0.014) | (0.023) |
| FS |  | 0.5388*** | 0.5300*** | 0.5257*** | 0.5274*** | 0.6107*** |
|  |  | (0.025) | (0.026) | (0.027) | (0.026) | (0.028) |
| FA |  | -0.0005 | -0.0003 | 0.0013 | 0.0001 | 0.0257 |
|  |  | (0.019) | (0.019) | (0.019) | (0.018) | (0.017) |
| DA |  | -0.0148 | -0.0171* | -0.0195* | -0.0193* | -0.0288*** |
|  |  | (0.010) | (0.010) | (0.010) | (0.010) | (0.011) |
| ROA |  | 1.2877*** | 1.2892*** | 1.2995*** | 1.2985*** | 1.3169*** |
|  |  | (0.083) | (0.082) | (0.080) | (0.079) | (0.089) |
| EMP |  | 0.1585*** | 0.1617*** | 0.1598*** | 0.1591*** | -0.3235*** |
|  |  | (0.021) | (0.020) | (0.020) | (0.020) | (0.023) |
| _cons | 9.2114*** | -4.0433*** | -3.8751*** | -3.7758*** | -3.8046*** | -4.4898*** |
|  | (0.007) | (0.456) | (0.464) | (0.490) | (0.464) | (0.484) |
| Year FE | No | Yes | Yes | Yes | Yes | Yes |
| Provincial FE | No | No | Yes | Yes | Yes | Yes |
| City FE | No | No | No | Yes | Yes | Yes |
| Regional FE | No | No | No | No | Yes | Yes |
| Industrial FE | No | No | No | No | Yes | Yes |
| Obs. | 33467 | 33467 | 33467 | 33467 | 33467 | 33467 |
| R-squared | 0.0261 | 0.9108 | 0.9121 | 0.9155 | 0.9156 | 0.8517 |

Notes: Standard errors are clustered at the city level and reported in parentheses

* $p < 0.10$

** $p < 0.05$

*** $p < 0.01$. The firm-level TFP in Columns (1)–(4) were estimated using the LP method, while the outcome variable in the last column was measured using the OP method.

(5) with the fixed effects of the control variables. The green finance pilot policy was an exogenous shock to the firms since a conclusion can be drawn that a significant causal relationship existed between green finance and a firm's TFP. The estimated coefficient of treated*post in Column 6 was 0.0622, which suggested that the implementation of the green finance pilot policy has stimulated the TFP of firms in pilot zones by approximately 6.22% compared with the firms in non-pilot zones. To this end, it could be concluded that green finance played a crucial role in firms' TFP stimulation, thereby verifying the effectiveness of the Porter Hypothesis (Hypothesis 1).

The sign of the selected control variables was consistent with this study's expectations. The LR, FS, FA, ROA and EMP coefficients positively affected firms' TFP. Theoretically, if a firm shows a high growth rate of over several periods, it will command multiples that exceed the current market multiple. Therefore, its stock price should increase when its forward multiple increases, resulting in a higher return for investors. However, RG exerted a negative effect on TFP. A possible explanation for this is that one crucial driver of higher productivity is process enhancement because of experience accumulation, also known as the so-called "learning-by-

doing" (The process of "learning-by-doing" may take several years to acquire in the case of young firms or start-ups). The Penrose Effect illustrates this challenge regarding managerial capabilities and the technology absorptive capacity as the key binding constraint that can hinder firms' growth [84]. To this end, firms may experience operational inefficiency following a period of uncontrolled rapid expansion. This situation is because of their inability to adjust managerial and other resources in time to deal with the additional organisational volatilities, uncertainties and complexities typically associated with a period of rapid firm growth [85]. The negative correlation between CEO duality and TFP designated that firms should decentralise their power to ensure multiple participants to participate in the decision-making processes.

## 4.2 Parallel trend analysis

The above benchmark regression confirmed the assumption that green finance can stimulate firms' TFP. However, there remained a lingering issue needing to be solved: are there any other potentially influential factors influencing the dependent variables? Subsequently, this study adopted the parallel trend analysis to ensure a common trend of firm-level TFP in the treatment and control groups before implementing the green finance pilot policy.

Referring to [69, 86], this study generated a series of year dummy variables tracking the effect of the CFRI pilot policy in 2017. The specific equation is denoted as follows:

$$y_{it} = \alpha_0 + \sum_{s=1}^{5} \alpha_{Pre\_s} did_{pre\_s} + \alpha_{current} did_{current} + \sum_{s=1}^{4} \alpha_{post\_s} did_{post\_s} + \beta \sum X_{it} + \varepsilon_{it} \qquad (8)$$

Where $did_{pre_s}$, $did_{current}$ and $did_{post_s}$ represent a series of year dummy variables tracking the impact of the green finance pilot policy in 2017. For example, because 2012 is in the first five years before the policy announcement, the variable $did_{pre_{-5}} = 1$, and for the other years, $did_{pre_{-5}} = 0$; because 2017 is the year of the policy announcement, then $did_{current} = 1$, and for the other years, it is equal to 0; because 2018 is the year after policy announcement, $did_{post\_(+1)} = 1$, and for the other years, it equals 0. While other information is consistent with Eq (1).

Fig 3 displays the common trend for firms' TFP from 2012 to 2021. The y-axis represents the effect of implementing the green finance pilot policy on the firms' TFP, and the x-axis denotes the time trend. Before the green finance pilot policy was implemented in 2017, the estimated coefficient on the dummy variable denoting the pilot policy implementation was not statistically significant. This result implied no difference between the treatment group and control groups before the green finance pilot, which confirmed the parallel trend assumption. Furthermore, to enhance the robustness of the results, this study also carried out an alternative parallel trend test following [87]. The estimated F-value equaled 1.83 (p-value = 0.1197), which did not reject the null hypothesis, thus, consolidating the above parallel trend results.

## 4.3 Placebo test

This study conducted a placebo test to check the robustness of the benchmark regression results. Subsequently, a placebo treatment was generated where the treatment group was randomly assigned to the control group or vice versa. The benchmark regression model was then re-run 1000 times. According to Fig 4, the mean value of the coefficient of *did* estimator is close to 0. Still, most of the corresponding p-values were greater than 10%, which suggested that random correlation across explanatory variables could not explain the estimated results.

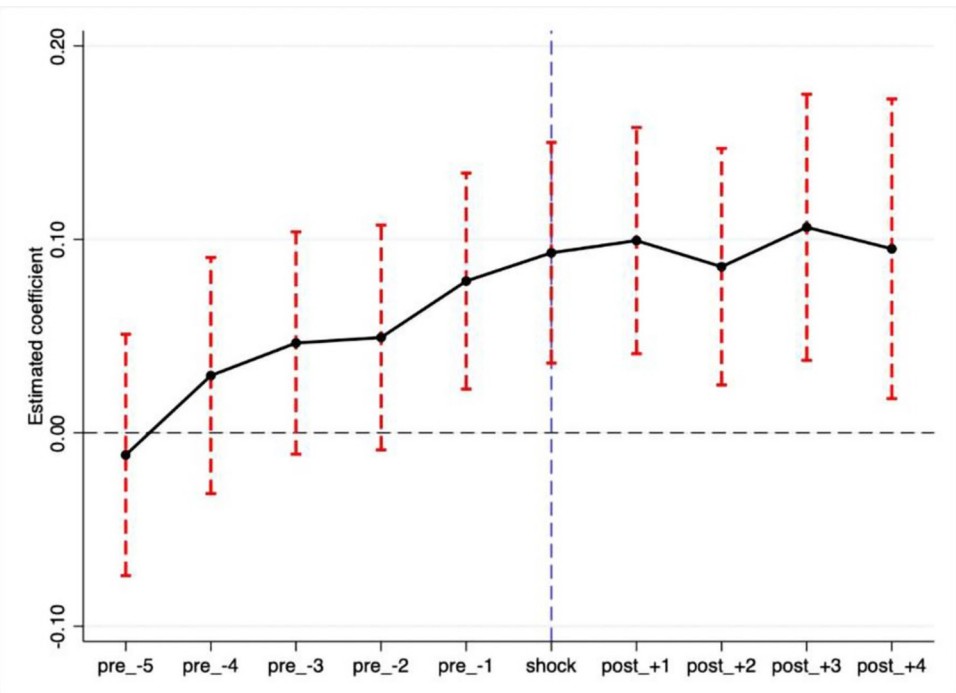

**Fig 3. Parallel trend test.** Notes: It is also noteworthy to mention that the "anticipation effect" (the estimated coefficient of the policy dummy variable pre_-1 was statistically significant in Fig 3) in this paper was also identified, which was consistent with [88]. In other words, firms may vigorously formulate strategic plans to respond to policy shock in advance.

## 4.4 PSM-DID and other robustness checks

Given that the selection of pilot regions for the CFRI policy in China may not be random, it is natural to have doubts about the randomness or whether there has truly been selection bias, thereby resulting in a biased estimation. Thus, this study then adopted difference-in-differences based on propensity score matching (PSM-DID) to verify further the estimated results' reliability [89]. Specifically, the logit model was constructed with the selected control variables (i.e., LR, RG, FS, FA, DA, ROA, and EMP) as matching variables. The K-nearest neighbour and kernel matching methods were selected to match the sample.

As reported in Table 4, the balance test results showed that there is no significant difference between the treatment and control groups after the matching, and the balanced datasets were then used to re-estimate the benchmark regression. Columns (1) & (2) in Table 4 present the corresponding results. It can be seen that the estimated coefficients of *did* in all models were statistically significant and positive no matter which matching methods were used.

Albeit with the above robustness test methods, Column (3) in Table 4 drops the firms in the year before the green finance pilot policy came into impact to avoid the so-called "anticipation effect" [88]. Column (4) in Table 4 moves the green finance pilot policy implementing time forward for three years to exclude the ex-ante effect. Still, Columns (5) & (6) displays the regression results by adopting the new method to re-estimate firm-level TFP, namely, the GMM approach [90] and ACF approach [91]. The estimated results were very robust. Thus, it is believed that implementing of the green finance pilot policy promoted firms' TFP stimulation in China.

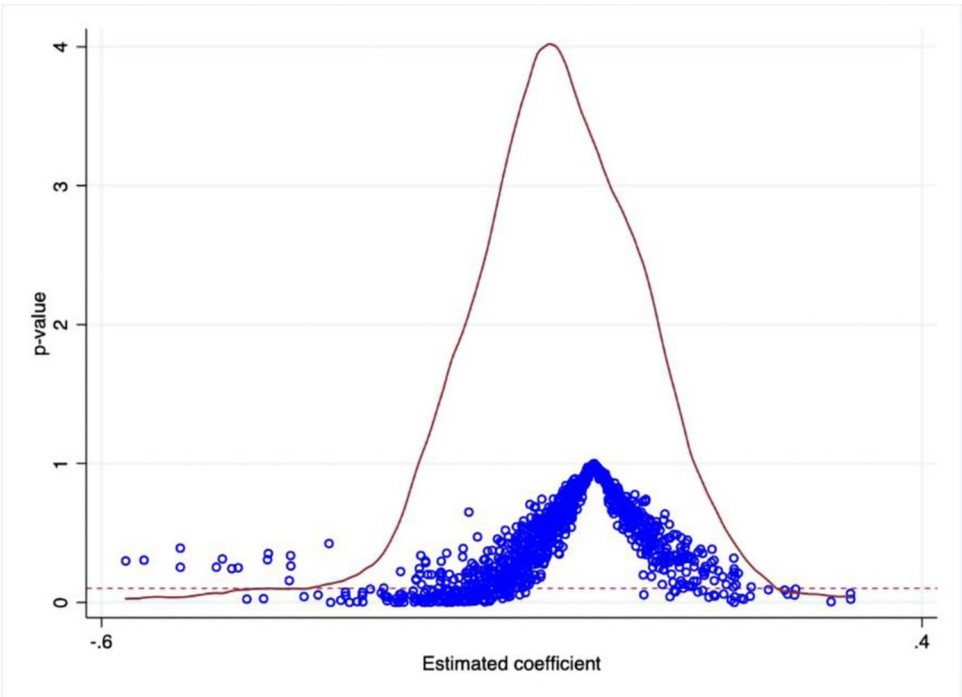

**Fig 4. Placebo test.**

## 5. Heterogeneity analysis

The Ref. [41] noticed that different types of firms differed in; dealing with risks, improving resource management and allocation, and responding to policy shocks. Therefore, the firms'

**Table 4. Robustness test results.**

| | PSM-DID | | Avoiding anticipation effect | Changing event window | TFP_GMM | TFP_ACF |
|---|---|---|---|---|---|---|
| | K-Nearest neighbour matching | Kernel matching | | | | |
| *did* | 0.0621*** | 0.0624*** | 0.0809*** | 0.0325*** | 0.0643*** | 0.0518*** |
| | (0.014) | (0.014) | (0.017) | (0.0112) | (0.015) | (0.015) |
| _cons | -3.8039*** | -3.7973*** | -3.6915*** | -4.5016*** | -3.4801*** | -3.8644*** |
| | (0.464) | (0.464) | (0.475) | (0.7000) | (0.485) | (0.513) |
| Control variables | Yes | Yes | Yes | Yes | Yes | Yes |
| Year FE | Yes | Yes | Yes | Yes | Yes | Yes |
| Provincial FE | Yes | Yes | Yes | Yes | Yes | Yes |
| City FE | Yes | Yes | Yes | Yes | Yes | Yes |
| Regional FE | Yes | Yes | Yes | Yes | Yes | Yes |
| Industrial FE | Yes | Yes | Yes | Yes | Yes | Yes |
| Obs. | 33461 | 33449 | 30082 | 15913 | 33467 | 33467 |
| R-squared | 0.9156 | 0.9156 | 0.9140 | 0.9457 | 0.8053 | 0.8631 |

Notes: Standard errors are clustered at the city level and reported in parentheses

* $p < 0.10$

** $p < 0.05$

*** $p < 0.01$. The firm-level TFP in Columns 1–4 was estimated using the LP method, while the outcome variable in the last column was measured using the GMM method.

decisions may have heterogeneity when responding to the shocks of financial institutions or banks from the green finance pilot policy. The first heterogeneity analysis focused on the nature of the firms' ownership. Chinese firms have state ownership characteristics. Unlike non-SOEs, the SOEs enjoy government guarantees and financing convenience and undertake more social responsibilities [92]. Thus, it is relatively easy to obtain additional government support for them. Given that the government may prefer to prioritise developing SOEs, they generally have softer environmental supervisions and regulations, causing them to face a lower "compliance cost".

Additionally, the financial institutions and banking system, characterised by state-owned banks, make bank supervision of SOEs less effective, which made it easier for SOEs to acquire financial resources. Therefore, under the green finance pilot policy shocks, SOEs may have stronger survival pressure than non-SOEs [60]. Considering [93], their whole sample was classified into two groups according to their ultimate controllers: the SOE group and the non-SOE group. As reported in Columns (1) & (2) of Table 5, the green finance pilot policy had a weak promoting effect on firms' TFP in the SEO group, while it had a significant positive effect in the non-SEO group, thus, confirming Hypothesis 2. The results showed that under the strict environmental regulations, non-SOEs were more sensitive to the green finance pilot policy shock because of the greater survival pressure.

China can be seen as a good case regarding regional heterogeneity. It is acknowledged that eastern China has taken the lead in developing its economy because of earlier policy preferences and its resource endowments. It can be inferred that the eastern region is well developed concerning; economic strength, technological innovation, and financial resources when compared with the central and western regions. Subsequently, this study divided China into three economic zones: Eastern China, Central China, and Western China, and we then perform the benchmark regression model were the performed to capture the potential regional heterogeneity. According to Table 5, the coefficients of *did* in all models were positive, but the green finance pilot policy only worked in the eastern and western regions, with a weak promoting effect in the western region. Thus, Hypothesis 3 was supported. Although the Chinese

**Table 5. Heterogeneity analysis.**

| | Ownership | | Region | | | Growth type | | |
|---|---|---|---|---|---|---|---|---|
| | SOEs | Non-SOEs | Eastern | Central | Western | Growth | Mature | Decline |
| *did* | 0.0358 | 0.0648*** | 0.0656*** | 0.0766* | 0.0485 | 0.0609*** | 0.0390 | 0.0139 |
| | (0.030) | (0.014) | (0.015) | (0.039) | (0.035) | (0.016) | (0.024) | (0.034) |
| _cons | -2.9170*** | -4.3840*** | -4.1226*** | -3.9466*** | -3.4756*** | -4.7099*** | -2.9806*** | -4.0218*** |
| | (0.892) | (0.613) | (0.493) | (1.127) | (1.146) | (0.413) | (0.704) | (0.756) |
| Control variables | Yes | Yes | Yes | Yes | Yes | Yes | Yes | Yes |
| Year FE | Yes | Yes | Yes | Yes | Yes | Yes | Yes | Yes |
| Provincial FE | Yes | Yes | Yes | Yes | Yes | Yes | Yes | Yes |
| City FE | Yes | Yes | Yes | Yes | Yes | Yes | Yes | Yes |
| Regional FE | Yes | Yes | Yes | Yes | Yes | Yes | Yes | Yes |
| Industrial FE | Yes | Yes | Yes | Yes | Yes | Yes | Yes | Yes |
| Obs. | 11282 | 22127 | 23586 | 5337 | 4514 | 16468 | 7344 | 7404 |
| R-squared | 0.9391 | 0.8875 | 0.9229 | 0.9179 | 0.8965 | 0.9125 | 0.9569 | 0.9562 |

Notes: Standard errors are clustered at the city level and reported in parentheses

* $p < 0.10$

** $p < 0.05$

*** $p < 0.01$. The firm-level TFP in all columns was estimated using the LP method.

government has implemented several powerful initiatives to support the western region's development (i.e., The Development of the Western Region) in recent years, the financial resources in the western region remain quite limited.

Additionally, the maturity of the financial system and mechanisms has been relatively low in the western region. At the same time, due to the lack of technical support and resource endowments, the western region's economic efficiency and resource allocation rate remain relatively low because many firms and industrial sectors have not been rearticulated successfully. In this regard, both local and central governments should pay more attention to the western region's development by providing sufficient financial support.

After considering the regional heterogeneity, this study examined firms' heterogeneity from the temporal dimension. That is, the evolution of a firm's life cycle varies across times. Specifically, the life cycle theory postulates that the levels of; financial constraints, financing and investment strategies, managerial behaviour, and the innovation willingness of firms during different development stages, are quite different [94]. Consequently, after firms systematically evaluate their; development demands, resource endowments, and external environment, their green innovation asks for prudent decision-making. Therefore, it is reasonable to infer that the incentive impacts of the green finance pilot policy on firms' TFP may vary depending on the stage of the life cycle that the firms are in, within a given period. Given this, it was also necessary for the present study to further explore the heterogeneity in terms of the firms' life cycles. A comprehensive index was calculated by the entropy method based on five variables: revenue growth rate, capital expenditure rate, firm age, retained earnings rate and the ROA to determine firm's life cycle stage.

Given that Chinese listed firms have moved past the start-up stage and considering the differences between industries, a firm's life cycle in this paper was classified into three stages: growth, maturity, and decline. As reported in Table 5, the estimated coefficient of *did* in Column (7) was statistically positive at a 1% significance level for firms in the growth stage, whereas the corresponding coefficients of *did* in Columns (8) & (9) were not significant for either the mature or declining stages, thereby rejecting Hypothesis 4. Besides, the magnitude was also slightly lower compared with firms in the growth stage. The estimated results revealed that implementing the green finance pilot policy was more effective in stimulating access to greater capital and information for firms in the growth stage. Thus, reducing the firms' financial constraints, raising managers' environmental protection awareness, and stimulating technological innovation capability, thus, ultimately improving the firms' TFP. This study also believes that firms with growth tend to be active because they want to occupy the market by providing high-quality and differentiated products, forcing them to conduct R&D activities.

## 6. Mechanism analysis

As a "surplus" in total output growth, TFP reflects the efficiency with which factor inputs are transformed into outputs, but it cannot be directly explained by factor inputs alone [77]. In addition to technological progress, TFP reflects the; knowledge level, managerial skills, institutional environment and so on, and is, therefore, more appropriately referred to as the productivity level. This section discusses the potential mechanisms by which green finance pilot policies can contribute to TFP at the firm level by influencing technological innovation and resource allocation.

### 6.1 Impact of technological innovation

This study used the ratio of firm R&D investment to sales (***TI1***) as a proxy variable to measure the firm-level technological innovation level to examine whether the green finance pilot policy

**Table 6. Mechanism analysis.**

| | | | | | | | | | | | |
|---|---|---|---|---|---|---|---|---|---|---|---|
| DID | 0.2738*** | | | DID | 0.4788*** | | | DID | -0.0676*** | | |
| | (0.023) | | | | (0.028) | | | | (0.010) | | |
| | | TI1 | 0.2630*** | | | TI2 | 0.1414*** | | | | |
| | | | (0.044) | | | | (0.023) | | | Efficiency | -0.9641*** |
| | | | | | | | | | | | (0.220) |
| _cons | 3.5527*** | _cons | -4.8810*** | _cons | 7.1920*** | _cons | -4.8238*** | _cons | -0.4313*** | | -4.2799*** |
| | (0.186) | | (0.184) | | (0.218) | | (0.187) | | (0.073) | | (0.141) |
| Control variables | Yes | | Yes | | Yes | | Yes | | Yes | | Yes |
| Year FE | Yes | | Yes | | Yes | | Yes | | Yes | | Yes |
| Provincial FE | Yes | | Yes | | Yes | | Yes | | Yes | | Yes |
| City FE | Yes | | Yes | | Yes | | Yes | | Yes | | Yes |
| Regional FE | Yes | | Yes | | Yes | | Yes | | Yes | | Yes |
| Industrial FE | Yes | | Yes | | Yes | | Yes | | Yes | | Yes |
| Obs. | 28825 | | 28825 | | 30142 | | 30142 | | 32338 | | 32338 |
| R-squared | 0.1012 | | 0.6261 | | 0.2711 | | 0.692 | | 0.6289 | | 0.4966 |

Notes: Standard errors are in parentheses

* $p < 0.10$

** $p < 0.05$

*** $p < 0.01$. The dependent variables in Columns 2, 6 & 10 were the selected proxy variables to measure firm-level technological innovation (i.e., **TI1** and **TI2**, respectively). While the dependent variable in Columns 4, 10 & 12 was firm-level TFP that is estimated using the LP method.

promotes firm-level TFP through the channel of technological innovation, following [95]. The ratio of R&D personnel to the number of employees (**TI2**) was also selected to enhance the robustness of the results. Referring to [96], this study first checked the correlation between green finance and firm innovation. Then the correlation between firm innovation and firms' TFP was checked in the second stage. The specific regression model was regulated as follows:

$$R\&D_{it} = \alpha_0 + \alpha_1 did_{it} + \beta \sum X_{it} + \varepsilon_{it} \tag{9}$$

$$TFP_{it} = \alpha_0 + \alpha_1 R\&D_{it} + \beta \sum X_{it} + \varepsilon_{it} \tag{10}$$

Where $R\&D_{it}$ was the proxy variable for firm innovation; the other information is consistent with Eq (1).

Columns (2), (4), (6) and (8) in Table 6 displays the corresponding empirical results. More specifically, Columns (2) & (6) were the regression results corresponding to Eq (9). The coefficients of *did* were statistically significant at the 1% level, which indicates that the green finance pilot policy also has significantly promoted firm innovation. The regression results in Columns (4) & (8) were estimated by regressing Eq (10). It can be seen that the corresponding coefficient of technological innovation was still also statistically positive at a 1% significance level. The above results demonstrated the effectiveness of the neoclassical theory that technological innovation plays a crucial role in firms' TFP improvement. To this end, it can be concluded that the green finance pilot policy also promoted firms' TFP through the channel of technological innovation. Therefore, Hypothesis 4 was reliable.

## 6.2 Impact of resource allocation efficiency

Resource allocation efficiency mainly regulates the consistency between marginal cost and the marginal cost in the process of allocating production factors across heterogeneous production

departments [97]. It is a result of many crucial factors, such as; market developments, technological progress, economic structure changes, control policies, macroeconomic management etc. [98]. Two distinct levels are used for measuring resource allocation efficiency, namely, the macro and micro levels. The present paper explored the effect of the green finance pilot policy on firm-level TFP by improving resource allocation efficiency at the micro level.

There are two distinct measurement schemes used for measuring resource allocation efficiency. One is to gauge the resource allocation efficiency from the perspective of a firm's ability to seize investment opportunities (The studies measured the firm-level resource allocation efficiency based on the investment-investment opportunity sensitivity model) [99, 100], while the second measure is to measure resource allocation efficiency from the investment efficiency perspective [101].

After entering the new era, the concept of quality development has become one of the most important guidelines for China to cultivate new growth drivers, further develop new growth models and achieve sustainable and inclusive economic development. As part of its efforts to promote sustainable economic development, the proposition of the supply-side structural reform emphasises the deleveraging policy to reduce excessive reliance on government, household, and firm finance borrowings. While the introduction of the supply-side structural reform policy plays a crucial role in Chinese firms' financing behaviour, such policy reform aims to solve overcapacity issues and further improve investment efficiency [102]. In this regard [103], has clearly demonstrated that whether a firm's investment grade matches its investment opportunities was an essential measure of resource allocation efficiency. Thus, to better capture the mediating effect of resource allocation efficiency in the incentive role of green finance pilot policy on firms' TFP stimulation, this study measured firms' resource allocation efficiency from the perspective of investment efficiency. Implementing a green finance pilot policy can enhance the firm-level investment efficiency, implying that the resource allocation rate in the selected firms is significantly enhanced. The improved efficiency utilisation is positively correlated with firms' TFP. Thus, this study demonstrated the promoting effect of the green finance pilot policy.

The neoclassical theory points out that firms are willing to invest when the marginal benefit equals the marginal cost of their investment to maximise their benefits [104, 105]. For simplicity, whether a firm's investment grade deviates from the so-called optimal level. Therefore, the matching consistency between a firm's actual capital investment grade and investment opportunity can be seen as a crucial indicator to measure the firms' resource allocation efficiency. Referring to [106], this study first estimated the firms' normal capital investment degree. The residuals from the regression model can reflect the deviation degree from the expected investment level, and, therefore, this study treated the absolute value of the residual as a firm-specific proxy variable for firms' resource allocation efficiency. The lower the value is, the higher the resource allocation will be. Accordingly, the model was specified as follows:

$$investment_{it} = \alpha_0 + \alpha_1 investment_{it-1} + \alpha_3 FS_{it-1} + \alpha_4 LR_{it-1} + \alpha_5 FA_{it-1}\alpha_6 RG_{it-1} + \alpha_7 SR_{it-1} + \alpha_8 cashholding_{it-1} + \varepsilon_{it}(11)$$

Where $investment_{it}$ represents the firms' investment level; $cashholding$ denotes the firms' cash holdings; $SR_{it-1}$ is the market-adjusted stock return. Other information was consistent with Eq (1). This study also controlled for the; year provincial, city, regional, and industrial fixed effects when estimating Eq (11) to ensure the consistency of the estimated results.

Consistent with the former mechanism analysis, a two-stage regression model is constructed to model the role of resource allocation efficiency between green finance and firms'

TFP. The model was stipulated as follows:

$$Efficiency_{it} = \alpha_0 + \alpha_1 did_{it} + \beta \sum X_{it} + \varepsilon_{it} \tag{12}$$

$$TFP_{it} = \alpha_0 + \alpha_1 Efficiency_{it} + \beta \sum X_{it} + \varepsilon_{it} \tag{13}$$

Where $Efficiency_{it}$ denotes the firms' resource allocation efficiency, and other information was consistent with Eq (1).

Columns (10) & (12) in Table 6 present the corresponding results. The coefficient of *did* in Column (10) was significantly negative, at the 1% significance level. In contrast, the coefficient of *Efficiency* in Column (12) was also statistically negative, at the 1% significance level. The results implied that China's green finance pilot policy had successfully improved firms' resource allocation efficiency, revealing that green finance exerted a certain effect on firms' TFP improvement through the lens of improving firms' resource allocation efficiency.

## 7. Conclusions and policy implications

This paper had shed new light on investigating the effect of the implementation of the green finance pilot policy on firms' TFP stimulation to fulfil the research gaps by implementing a quasi-natural experiment method, spanning from 2012 to 2021. The main conclusions of this paper are given as follows. Firstly, the benchmark regression results showed that implementing China's green finance pilot policy significantly improved firms' TFP. The estimated benchmark regression results were very robust after a series of robustness tests. No matter which robustness tests are used, such as; the placebo test, PSM-DID (i.e., K-nearest neighbouring matching method and kernel matching method), excluding the ex-ante effect, changing the event window, and replacing the measurement methods of the explained variables (i.e., GMM approach and ACF approach). Secondly, the heterogeneity analysis results suggested that the effect of the green finance pilot policy was prominent for non-SOEs, firms in eastern China and central China, and firms in growth type firms, while it had a relatively weak effect on SOEs, western China, and firms in mature and declining types firms. Finally, the green finance pilot policy could also stimulate firms' TFP growth through technological innovation and resource allocation efficiency.

Based on the derived empirical findings, several policy implications have been proposed. Firstly, the empirical results showed that implementing a green finance pilot policy positively affected firms' TFP. Hence, the government should continue the pilot policy and improve it, thereby increasing investment in green financial facilities and strengthening financial institutions' and firms' environmental risk awareness. At the same time, the government should change the intrinsic thought of a one-size-fits-all strategy and formulate the strategic green finance pilot policy by taking various heterogeneities (i.e., firms' characteristics, regional differences, and firms' life cycles) into consideration.

Secondly, policymakers should reinforce the development of the national financial, economic, and social conditions to enhance further the technological innovation capability and absorptive capacity through green finance initiatives. By raising the financial threshold (i.e., improving interest rates) for those firms with high energy consumption and high pollutant emissions, the "zombie enterprises" will be phased out of the market, thereby consolidating the survival of the fittest mechanism. In the meantime, under the stringent environmental regulations, firms must resile self-transformation and self-adjustment by sharing the budget to invest in more R&D activities, especially green technologies. In contrast, adopting environmentally-friendly production equipment and techniques is expected to improve economic efficiency and reduce pollution emissions, thus, realising a "win-win" situation.

Thirdly, it is recommended that China should strongly promote the supply-side structural reform policy to improve resource allocation efficiency further. To this end, the adjustment and transformation of the industrial structure becomes more urgent. This situation is because transforming traditional industries and guiding firms to gradually shift from relying solely on resource and cost advantages to relying on green technology innovation will cultivate new growth momentums. The advancement of technological innovation typically attained by large firms can be a case to sustain the green transformation of small- and medium-sized firms. To this end, it is also necessary to realise the national overall industrial adjustment from "terminal governance" to "cleaner production". In addition, it is also suggested that China should develop high-tech manufacturing industries, optimise its industrial structure, and improve resource allocation efficiency through the development and utilisation of clean energy (i.e., hydropower, nuclear).

## Supporting information

**S1 Appendix. List of covered pilot regions.**
(DOCX)

## Acknowledgments

We would like to show our greatest appreciation to the anonymous reviewers, editor, and those who have helped to contribute to this paper writing.

## Author Contributions

**Conceptualization:** Miaomiao Tao.

**Formal analysis:** Hanghang Dong, Miaomiao Tao.

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
