## [Decision Letter · Decision Letter 0]

22 Aug 2022

PONE-D-22-16731Does China’s green finance policy stimulate firms’ total factor productivity? An empirical evidence based on China’s pilot zones for green finance reform and innovationsPLOS ONE

Dear Dr. Dong,

Thank you for submitting your manuscript to PLOS ONE. After careful consideration, we feel that it has merit but does not fully meet PLOS ONE’s publication criteria as it currently stands. Therefore, we invite you to submit a revised version of the manuscript that addresses the points raised during the review process.

We look forward to receiving your revised manuscript.

Kind regards,

László Vasa, PhD

Academic Editor

PLOS ONE

Journal Requirements:

3. In your Methods section, please include additional information to report the original source of the data and the methods used to collect it in sufficient detail for another researcher to access the same data. Please also ensure that you have included a statement specifying whether the collection and analysis method complied with the terms and conditions for the source of the data.

6. Thank you for stating the following financial disclosure:

“no”

7. Thank you for stating the following in your Competing Interests section: 

“NO”

8. Thank you for submitting the above manuscript to PLOS ONE. During our internal evaluation of the manuscript, we found significant text overlap between your submission and the following previously published works.

- https://www.sciencedirect.com/science/article/abs/pii/S0040162522001901?via%3Dihub

- https://www.sciencedirect.com/science/article/abs/pii/S0040162522000488?via%3Dihub

- https://link.springer.com/article/10.1007/s11187-014-9584-2

- https://www.sciencedirect.com/science/article/pii/S0140988322001645?via%3Dihub

- https://www.sciencedirect.com/science/article/abs/pii/S095965262200782X?via%3Dihub

- https://www.sciencedirect.com/science/article/pii/S0160791X22000471?via%3Dihub

Please revise the manuscript to rephrase the duplicated text, cite your sources, and provide details as to how the current manuscript advances on previous work. Please note that further consideration is dependent on the submission of a manuscript that addresses these concerns about the overlap in text with published work.

Reviewers' comments:

Reviewer's Responses to Questions

**Comments to the Author**

1. Is the manuscript technically sound, and do the data support the conclusions?

Reviewer #1: Partly

Reviewer #2: Yes

2. Has the statistical analysis been performed appropriately and rigorously? 

Reviewer #1: No

Reviewer #2: Yes

3. Have the authors made all data underlying the findings in their manuscript fully available?

Reviewer #1: No

Reviewer #2: Yes

4. Is the manuscript presented in an intelligible fashion and written in standard English?

Reviewer #1: No

Reviewer #2: Yes

5. Review Comments to the Author

Reviewer #1: Dear Authors,

The paper titled “Does China’s green finance policy stimulate firms’ total factor productivity? An empirical evidence based on China’s pilot zones for green finance reform and innovations” targets to analyze the differential and heterogeneity effect of TFP on firm level in China. I have read the complete paper and I hold following observation on it.

The title is bit long, it duplicates similar terms in both parts. Abstract is good and covers important feature of the manuscript. Introduction is well written. However, the debate of how this study is different from existing studies is quite long, which can be written precisely and have a better understanding for the reader. As in third paragraph where authors already stressed why this study is important. In subsequent paragraphs, the authors started again a debate on why green finance got widespread attention globally. However, the authors must pave the ground of stating the importance of study and how it is novel in context of existing ones. The topic seemed good, but there is lack of coherence in stressing the need of this study. Authors may consider restructuring introduction part and highlight the novelty of this study in coherent way. Moreover, the writeup requires a careful proofread as I found several misspelled words e.g., para 4 (word sustian), para 5 (ablite, intention). Some of the statements are not clear e.g., para 5 (…. but ignored the role of green finance) role of green finance in what? Appropriate citation should be provided in introduction part where authors mentioned about “executive conference of the State Council” and “Porter’s hypothesis”.

The literature review is written well. However, except explaining concepts of green finance, institutional background, and environmental constraints, authors should provide literature from existing studies concerning green finance and TFP. There are several studies that commented on the impact of green finance and TFP of firms, which can be included in the manuscript to formulate hypotheses. Furthermore, hypotheses 2 and 3 are not clear. What does “more obvious” mean? Authors need to address this issue and reformulate these hypotheses based on theory and existing empirical studies. There is serious problem of coherence in the write up, for instance, para 3 of hypothesis formulation. Authors are contradicting the previous statement in every next sentence. I would recommend a proofread of the manuscript.

Methodology part is well developed. However, methodology must provide details of all 5 pilot zones that implemented GFRI. Also, what other zones were considered in the study as control group? In DID, the prime focus is to be given in selection of control and treatment group. The selection of untreated group is of profound importance, and there are higher chances of selecting inappropriate control group that does not show the similar trajectory outcome variables (Abadie and Gardeazabal, 2003). The selection criteria should also be mentioned. Furthermore, as mentioned by authors, the time of intervention is 2017 in pilot zones in China, but the question is when did the firms start implementing the GFRI intervention? Another point, I would suggest providing results of the models where you included and excluded the control variables. The difference is not visible.

The discussion and conclusion parts are good. However, I have observed that author made longer and over complicated sentences which affect the understandability and cohesiveness of the manuscript. It can be improved.

Reviewer #2: The study presents an interesting and important topic. The literature of the manuscript is outstanding, and the number of references is at the level expected of a scientific journal. The structure of the paper is appropriate, and the methodology and presentation of results are of the expected standard. What I would suggest to the authors, for the sake of better transparency, is that in the conclusions section, a summary table should be provided to summarise the evidence for the hypotheses. Also, please indicate the source and the method used under the tables and figures. I request that the authors edit the paper as required by the journal.

6. PLOS authors have the option to publish the peer review history of their article (what does this mean?). If published, this will include your full peer review and any attached files.

Reviewer #1: No

Reviewer #2: No

---

## [Author Response · Author response to Decision Letter 0]

30 Sep 2022

Dear Reviewers,

We thank the reviewers for their generous comments on the manuscript and have edited the manuscript to address their concerns.

Reviewer #1 raised several concerns, which we have carefully considered and made every effort to address. We fundamentally agree with all the comments made by the Reviewers, and we have incorporated corresponding revisions into the manuscript (Revised Manuscript with Track Changes).

Our detailed, point-by-point responses to the editorial and reviewer comments are given in the word file (Response to Reviewers), whereas the corresponding revisions are present in the manuscript file (Revised Manuscript with Track and Manuscript). Specifically, red text indicates changes made in response to the suggestions of Reviewer #1 and Reviewer #2. Additionally, we have carefully revised the manuscript to ensure that the text is optimally phrased and free from typographical and grammatical errors.

We believe that our manuscript has been considerably improved as a result of these revisions and hope that our revised manuscript "The policy effect of green finance reform and innovations: Empirical evidence at the firm level" is acceptable for publication in the PLOS ONE.

---

## [Decision Letter · Decision Letter 1]

10 Nov 2022

The policy effect of green finance reform and innovations: Empirical evidence at the firm level

PONE-D-22-16731R1

Dear Dr. Dong,

We’re pleased to inform you that your manuscript has been judged scientifically suitable for publication and will be formally accepted for publication once it meets all outstanding technical requirements.

Kind regards,

László Vasa, PhD

Academic Editor

PLOS ONE

Additional Editor Comments (optional):

Reviewers' comments:

Reviewer's Responses to Questions

**Comments to the Author**

1. If the authors have adequately addressed your comments raised in a previous round of review and you feel that this manuscript is now acceptable for publication, you may indicate that here to bypass the “Comments to the Author” section, enter your conflict of interest statement in the “Confidential to Editor” section, and submit your "Accept" recommendation.

Reviewer #3: All comments have been addressed

2. Is the manuscript technically sound, and do the data support the conclusions?

Reviewer #3: Yes

3. Has the statistical analysis been performed appropriately and rigorously? 

Reviewer #3: Yes

4. Have the authors made all data underlying the findings in their manuscript fully available?

Reviewer #3: Yes

5. Is the manuscript presented in an intelligible fashion and written in standard English?

Reviewer #3: Yes

6. Review Comments to the Author

Reviewer #3: The paper was revised along my instructions and now it is more sound and represents a higher scientific quality. I recommend it for publication without any further changes and improvements.

7. PLOS authors have the option to publish the peer review history of their article (what does this mean?). If published, this will include your full peer review and any attached files.

Reviewer #3: No

---

## [Editor Report · Acceptance letter]

15 Nov 2022

PONE-D-22-16731R1 

The policy effect of green finance reform and innovations: Empirical evidence at the firm level 

Dear Dr. DONG:

I'm pleased to inform you that your manuscript has been deemed suitable for publication in PLOS ONE. Congratulations! Your manuscript is now with our production department. 

Kind regards, 

on behalf of

Prof. Dr. László Vasa 

Academic Editor

PLOS ONE